# Non-Coding RNAs in Neurodevelopmental Disorders—From Diagnostic Biomarkers to Therapeutic Targets: A Systematic Review

**DOI:** 10.3390/biomedicines13081808

**Published:** 2025-07-24

**Authors:** Katerina Karaivazoglou, Christos Triantos, Ioanna Aggeletopoulou

**Affiliations:** 1Department of Psychiatry, University of Patras, 26504 Patras, Greece; karaivaz@hotmail.com; 2Division of Gastroenterology, Department of Internal Medicine, University of Patras, 26504 Patras, Greece; chtriantos@upatras.gr

**Keywords:** neurodevelopmental disorders, autism, ncRNAs, lncRNAs, miRNAs, biomarkers, gene regulation

## Abstract

**Background**: Neurodevelopmental disorders, including autism spectrum disorder (ASD) and attention-deficit/hyperactivity disorder (ADHD), are increasingly recognized as conditions arising from multifaceted interactions among genetic predisposition, environmental exposures, and epigenetic modifications. Among epigenetic mechanisms, non-coding RNAs (ncRNAs), including microRNAs (miRNAs), long non-coding RNAs (lncRNAs), and PIWI-interacting RNAs (piRNAs), have gained attention as pivotal regulators of gene expression during neurodevelopment. These RNA species do not encode proteins but modulate gene expression at transcriptional and post-transcriptional levels, thereby influencing neuronal differentiation, synaptogenesis, and plasticity. **Objectives**: This systematic review critically examines and synthesizes the most recent findings, particularly in the post-COVID transcriptomic research era, regarding the role of ncRNAs in the pathogenesis, diagnosis, and potential treatment of neurodevelopmental disorders. **Methods**: A comprehensive literature search was conducted to identify studies reporting on the expression profiles, functional implications, and clinical relevance of ncRNAs in neurodevelopmental disorders, across both human and animal models. **Results**: Here, we highlight that multiple classes of ncRNAs are differentially expressed in individuals with ASD and ADHD. Notably, specific miRNAs and lncRNAs demonstrate potential as diagnostic biomarkers with high sensitivity and specificity. Functional studies further reveal that ncRNAs actively contribute to pathogenic mechanisms by modulating neuronal gene networks. **Conclusions**: Emerging experimental data indicate that the exogenous administration of certain ncRNAs may reverse molecular and behavioral phenotypes, supporting their therapeutic promise. These findings broaden our understanding of neurodevelopmental regulation and open new avenues for personalized diagnostics and targeted interventions in clinical neuropsychiatry.

## 1. Introduction

Neurodevelopmental disorders are a group of early-onset, chronic conditions characterized by aberrant brain development that affects communication, social interaction, sensory processing, and behavior. There is a growing body of literature focusing on the pathophysiology of neurodevelopmental disorders, which has revealed a complex interaction between genetic and environmental factors [1]. According to the SFARI gene database, which is a constantly evolving database centered on genes implicated in autism susceptibility, more than 1000 genes have been implicated in the pathophysiology of autism and several genetic causes of neurodevelopmental disorders have been established including de novo copy number variants, single-nucleotide variants, short insertion and deletions (Indels), mosaic single-nucleotide variants and inherited recessive variants [1,2,3]. Rare de novo or inherited variants exert a large effect size on autism risk and are associated with 15–20% of ASD liability, while more than 50% of ASD liability is attributed to common variants, whose individual effect size is quite low. Common inherited variants are frequently found in the general population and subserve genetic diversity; however, the combination of certain variants may be associated with increased autism susceptibility risk [4,5]. For example, there is evidence from multiplex families’ studies that the presence of rare variants acts additively to increase autism genetic risk due to a combination of ASD associated common variants and may explain the complex genetic architecture of neurodevelopmental disorders [6].

Additionally, to genetic influences, environmental agents have been implicated in neurodevelopmental disorders’ pathophysiology. More specifically, several environmental exposures during gestation or neonatal life, such as maternal immune activation due to infections, stress, autoimmune or metabolic diseases, dietary factors, heavy metals, air pollutants, medications, and perinatal complications, have been associated with increased autism risk in offspring [7,8,9,10]. Research findings have repeatedly shown that these environmental influences, especially during intrauterine life, act through epigenetic mechanisms that modulate gene expression in genetically susceptible individuals, thus altering the course of neurodevelopmental processes [11]. In this context, research studies have begun to focus on the role of those epigenetic mechanisms in the emergence of neurodevelopmental deficits.

Non-coding RNAs are RNA molecules that do not encode proteins and regulate gene expression, thus affecting various cellular processes [12]. They are classified into two main categories, namely housekeeping ncRNAs, which are expressed in great amounts in all cells and are involved in generic cellular processes such as protein synthesis, RNA splicing, and RNA modification, and regulatory ncRNAs, which are tissue-specific and regulate gene expression. Housekeeping ncRNAs include ribosomal RNAs (rRNAs), transfer RNAs (tRNAs), small nuclear RNAs (snRNAs), small nucleolar RNAs (snoRNAs), TERC (telomerase RNAs), tRNA-Derived Fragments (tRFs), and tRNA halves (tiRNAs), while regulatory ncRNAS encompass micro RNAs (miRNAs), small interfering RNAs (siRNAs), PI-WI interacting RNAs (piRNAs), enhancer RNAs (eRNAs) circular RNAs (circRNAs), long non-coding RNAs (lncRNAs) and Y RNAs (YRNAs) [13,14,15,16]. There is a wide range of mechanisms through which non-coding RNAs act, for example, through chromatin remodeling and histone modification, transcription factors binding, mRNA degradation and translation inhibition, interaction with splicing factors and RNA-binding proteins [14]. Research findings have consistently shown that non-coding RNAs are implicated in the pathophysiology of a wide range of mental health conditions, including neurodegenerative disorders, mood disorders, schizophrenia, and neurodevelopmental disorders, and could potentially represent promising biomarkers and intervention targets [13,14].

During the present decade, following the COVID-19 pandemic, there have been major advancements in the field of molecular biology regarding the development and application of novel RNA-based diagnostics in all medical fields, including neuropsychiatry, leading to a rapidly expanding list of studies assessing the contribution of non-coding RNAs in neurodevelopmental outcomes [17]. In this context, the aim of the present systematic review was to detect and critically examine all literature data reporting on the role of non-coding RNAs in the pathophysiology of neurodevelopmental disorders, exclusively focusing on the last 5-year period (2020–2025). Earlier reviews on the topic [18,19,20] provide comprehensive and scientifically reliable knowledge regarding research conducted prior to 2020, while two more recent systematic reviews, although they screened literature till 2021 [21] and the early 2022 [22], they did not include all classes of ncRNAs, but instead focused on miRNAs and short ncRNAs, respectively. In this respect, due to the rising bulk of relevant information during the most recent years and the current use of more refined and accurate techniques of transcriptome analysis, an updated review is deemed necessary.

## 2. Materials and Methods

### 2.1. Search Strategy

We conducted a thorough systematic search of the literature across Pubmed and ScienceDirect using the following phrases, “non-coding RNAs and neurodevelopmental disorders” and “non-coding RNAs and autism”, in two consecutive searches. In addition, we conducted a manual search using all references of the selected articles to detect additional relevant research in the literature. The literature review was performed according to the guidelines from the Preferred Reporting Items for Systematic Reviews and Meta-analyses (PRISMA). Moreover, it has been registered in the Open Science Framework (OSF) under the following DOI: 10.17605/OSF.IO/93WKS. The last search was performed on 23 March 2025.

### 2.2. Eligibility Criteria

Inclusion criteria for the present review were (1) original articles published in English, (2) studies detecting non-coding RNAs in humans diagnosed with autism or any other neurodevelopmental disorders or in animal models of autism or other neurodevelopmental disorders, (3) studies published between 2020–2025. We also decided to exclude case reports and studies whose methodology was entirely based on computational methods and did not include any in vivo or in vitro experiments.

### 2.3. Study Selection and Data Extraction

Our search initially yielded 396 titles, but 61 titles were duplicates and were excluded, leaving 335 titles. In addition, 107 articles were reviews, and 3 articles had not yet been peer-reviewed (preprints) and were excluded from the study, leaving 225 articles. Two reviewers (KK and IA) independently reviewed all remaining abstracts according to the aforementioned inclusion and exclusion criteria. During the abstract review process, 4 articles were excluded due to being case reports, 2 articles were excluded because they were validation studies of ncRNAs detection methods, and 10 articles were excluded because they were based entirely on computational methods. The remaining 209 full texts were independently screened by three reviewers (KK, CT, IA), and 74 studies were excluded because they did not involve neurodevelopmental disorders, and 90 studies were excluded because they did not encompass the detection of non-coding RNAs, thus leaving 45 studies to be included in the current review. However, 1 study was retracted in March 2025, and finally, 44 studies entered the review process. Figure 1 contains a PRISMA flow diagram that illustrates the articles’ selection process.

## 3. Results

### 3.1. Studies’ Characteristics

Thirty-two (32) studies [23,24,25,26,27,28,29,30,31,32,33,34,35,36,37,38,39,40,41,42,43,44,45,46,47,48,49,50,51,52,53,54] included human samples, 11 studies [55,56,57,58,59,60,61,62,63,64] recruited animal models of neurodevelopmental disorders, and 1 study’s [65] methodology included both in vitro human and animal cells and in vivo laboratory animal models. 27 studies [24,25,30,32,33,34,36,37,38,39,40,41,42,47,48,49,50,52,53,54,55,57,58,59,61,63,65] focused on autism exclusively, while 2 studies included participants with autism and participants with other neurodevelopmental disorders, such as attention-deficit hyperactivity disorder (ADHD) [35] or intellectual disability [44]. In addition, 4 studies focused exclusively on ADHD [26,35,43,64], 3 studies [28,46,51] on Prader-Willi Syndrome, 2 studies [26,62] on Fetal Alcohol Spectrum Disorders and 2 studies [31,60] on learning disability or specific learning disorder, while the remaining 4 studies [23,29,45,56] involved subjects or animal models of neurodevelopmental disorder or delay. Likewise, 20 studies [27,31,33,35,36,37,38,39,42,43,46,48,49,50,54,55,57,60,64,65] assessed the association of lncRNAs with neurodevelopmental conditions, 10 studies [29,30,32,40,44,45,52,53,56,59] focused on miRNAs, 1 study [23] on small nuclear RNAs (snRNAs), 2 studies [28,51] on small nucleolar RNAs (snoRNAs), 2 studies [58,62] on circRNAs, 1 study [26] on small ribosomal RNAs, while in the remaining 8 studies [24,25,34,35,41,47,61,63] a combination of different ncRNAs was assessed. Given the heterogeneity of study designs, biological models, and reported outcomes, the findings are presented thematically according to the class of non-coding RNA and the associated neurodevelopmental condition. The characteristics of studies reporting on the role of non-coding RNAs in neurodevelopmental disorders are summarized in Table 1. Figure 2 illustrates the studies’ distribution regarding the neurodevelopmental condition and the type of ncRNA.

### 3.2. Long Non-Coding RNAs

LncRNAs constitute a sub-category of non-coding RNAs with a length that exceeds 200 nucleotides. These molecules have been implicated in various cellular processes, including proliferation, apoptosis, invasion, metastasis, and DNA damage, and play a crucial part in gene expression due to their ability to interact with proteins, lncRNA-protein complexes (lncRNP), and chromatin modifiers [38,66]. A large percentage of lncRNAs are specifically transcribed in the brain, thus modulating the expression of numerous protein-coding genes involved in neuronal growth, differentiation, and synaptogenesis [67]. Their mode of action encompasses a variety of mechanisms such as molecular signaling, scaffolding, decoying, guiding, organizing chromatin architecture, and acting as miRNA sponges. Molecular signaling involves, for example, the recruitment of chromatin modifiers that alter chromatin structure, thus repressing gene expression [68,69]. Scaffolding refers to the connection of protein complexes to facilitate their action, while decoying is the lncRNAs’ binding to proteins or other RNAs, preventing them from binding to their target sites [70]. Likewise, lncRNAs may guide transcriptional complexes to specific genomic locations, allowing control of chromatin modifications that regulate target genes’ expression or may act as architectural agents contributing to the spatial organization of the genome [71]. In addition, lncRNAs may bind to miRNAs, preventing them from mediating the degradation of mRNAs and the repression of mRNA translation [72]. Finally, although lncRNAS do not have the potential to be translated into full proteins, there is a small percentage of them that can be translated into micropeptides, which are involved in specific intracellular regulatory processes [66,73].

Several studies have shown that a large number of lncRNAS are differentially expressed in the hippocampus of animal models of autism, ADHD, and learning disability and are associated with the emergence of neurodevelopmental deficits [35,55,57,60,64]. In a similar way, numerous lncRNAs have been found to be either up- or down-regulated in several cortex areas and the striatum of an animal model of autism. Some of these lncRNAs have been shown to contribute to molecular pathways that are involved in neural migration, differentiation, and apoptosis [61]. For example, the lncRNA *NDIME*, which is the homologue of the human lncRNA *LINC00461*, was significantly downregulated in the hippocampus of a mouse model of autism. In addition, it was highly expressed in mice neural embryonic cells during neural development, and its knockdown downregulated the expression of several neural genes such as *Emx1*, *Brn2*, *Sox5*, and *Zeb2* and delayed neural differentiation [55]. NDIME was found to physically bind to EZH2, the major component of polycomb repressive complex 2 (PRC2), thus blocking the trimethylation of histone H3 lysine 27 (*H3K27me3*) at the *Mef2c* promoter and promoting *MEF2C* transcription. The knock-down of *NDIME* led to the downregulation of the *MEF2C* gene, which is considered to be involved in the pathogenesis of autism. According to the same study, the virus-mediated expression of *NDIME* in the hippocampus of the experimental models contributed to the improvement of autism-like symptoms [55]. Another research group showed that the knockdown of *LINC00461* is associated with cognitive deficits and impaired neural migration in mice [35].

According to previous research, *LINC00461* may act as a sponge to certain miRNAs (for example, the miRNA-411-5p), preventing them from binding to their target mRNAs [74]. In a similar way, the lncRNA *H19* was found to be increased in autism and to compete with the action of miR-484, which was downregulated in autistic probands. The miR-484 had a complementary strand of *H19*, and the latter targeted miR-484 and prevented it from acting on its mRNA targets [34]. In another study, the intraventricular injection of exosomal lncRNA *IFNG-AS1* to an animal model of autism led to increased levels of that lncRNA in prefrontal cortex neurons, ameliorated neuroinflammation, promoted neurogenesis in these brain areas, and improved autism-like social deficits. The lncRNA *IFNG-AS1* was shown to act as a molecular sponge for miR-21a-3p, regulating neurogenesis through the miR-21a-3p/PI3K/AKT axis [57]. Moreover, Mizuno et al. (2020) revealed that the lncRNA *Gm26917* is a key component of a huge RNA-RNA interaction network, which is involved in autism pathogenesis and acts by competing with miRNA-29b, which promotes cellular apoptosis [61]. Finally, in an earlier investigation, the lncRNA *NONRATT006598.2* was downregulated in an animal model of ADHD, while the administration of methylphenidate increased its expression levels. In addition, it was shown to play an important regulatory role in the hippocampal expression of *Baiap2*, which is involved in dendritic spine formation and is downregulated in neurodevelopmental disorders such as autism and hyperactivity syndromes [64].

Similarly to animal studies, human studies have revealed a variety of lncRNAs which are differentially expressed in the blood, neural, or pluripotent cells of individuals with autism, ADHD, or Prader-Willi syndrome [24,34,38,42,46,47,48,49,50]. In addition, structural variants of certain lncRNAs have also been associated with increased autism and ADHD risk [27,35,39,43]. Cheng et al. (2020) identified a panel of 20 non-coding RNAs, including two lncRNAS, the *AQP4-AS1* and the *MUC20-OT1*, which can discriminate between ASD cases and controls with high sensitivity and specificity [24]. Another group of lncRNAs, namely *LincRNA-ROR*, *LINC-PINT*, *LincRNA-p21*, *PCAT-29*, and *PCAT-1*, was also found to be downregulated in children with ASD compared to healthy controls, and *LincRNA-ROR* and *PCAT-1* exhibited high diagnostic power to detect ASD cases [42]. Furthermore, in a small-scale intervention study, the long ncRNA *MALAT-1*, which is involved in synaptogenesis and autism pathogenesis, was found to be significantly down-regulated after the implementation of a 1-year intensive cognitive-behavioral intervention program in ASD toddlers, highlighting the potential effect of psychosocial interventions on gene expression and molecular mechanisms [37]. However, these findings should be considered quite preliminary due to the extremely small sample size. In addition, according to a more recent study, multiple single nucleotide polymorphisms in lncRNAs including *LINC00461*, *PLK1S1-007* and *RP1-111D6.3*, were identified in ADHD individuals and were significantly associated with gray matter alterations of the superior/middle frontal regions [27], while rare structural variants have been identified in two non-coding RNAs genes, *LOC124905257* and *PTCHD1-AS*, in ASD individuals [35]. Moreover, Li et al. (2021) [33] revealed three lncRNAs, *DNM3OS*, *IGFBP7-AS1*, and *LINC01139*, which were differentially expressed in both neural progenitor cells and neurons in ASD patients, while *LINC00461* was identified as a risk gene for ADHD traits [35]. Likewise, the expression of the following lncRNAs, *CSNK1A1P*, *LRRC2-AS1* and *CCAT1* was reduced in ASD individuals [38,48,50], while the lncRNAs *DISC2*, *PRKAR2A-AS1*, *LOC101928237*, *CCAT2* and *MEG3* were upregulated in ASD patients and among these lncRNAs, *CSNK1A1P*, *CCAT2*, *LOC101928237*, *LRRC2*-*AS1* and *MEG3* exhibited high diagnostic accuracy in distinguishing between ASD cases and controls and could serve as biomarkers [48,49,50]. In addition, certain single-nucleotide polymorphisms of the lncRNA *HOTAIR* (HOX transcript antisense intergenic RNA) were found to be associated with increased ASD and ADHD risk [39,43]. Finally, an in vitro study showed that the knock-down of the *CHD8* (Chromodomain Helicase DNA-Binding protein 8) gene, which is an ASD risk gene, leads to the overexpression of the natural antisense lncRNA *RAB11B-AS1*, which increases endogenous RAB11B protein levels by enhancing gene translation [54]. In line with these findings, di Leva et al. (2025) [65] revealed that the administration of synthetic SINEUP-*CHD8* molecules, which are antisense lncRNAs able to stimulate the translation of sense target mRNAs, can increase Chd8 protein production in animal models and patients derived fibroblasts and revert the molecular and phenotypic effects of *CHD8* knock-down in animal models.

### 3.3. MicroRNAs

MiRNAs are single-stranded non-coding RNAs, 22 nucleotides long, highly conserved throughout evolution, that can modulate gene expression at the post-transcriptional level [13,75]. MiRNA genes are transcribed by RNA polymerase II (RNA polII) into primary miRNAs (pri-miRNAs), which are processed into precursor miRNAs (pre-miRNAs) and finally into mature miRNAs. These mature miRNAs are loaded into the RNA-induced silencing complex (RISC) and bind to complementary sequences on target messenger RNAs (mRNAs), thus leading to either mRNA degradation or inhibition of translation and the downregulation of targeted genes [47,76,77]. MiRNAs have been implicated in various neuropsychiatric disorders including Alzheimer’s disease (AD), Parkinson’s disease (PD), Huntington’s disease (HD), Fragile X syndrome (FXS), ASD, ADHD, Tourette syndrome (TS), schizophrenia (SCZ), and mood disorders and appear as promising candidate biomarkers for these conditions [13]. However, it should be noted that among the over 2600 human miRNAs that have been identified and registered in scientific databases, there is a large number of these non-coding molecules whose functional role has not yet been elucidated, and further information is needed regarding their contribution to normal human development and disease processes [13,78].

Animal studies have consistently shown altered levels of miRNAs in brain tissues and placentas derived from autism animal models [56,59,63]. Butler et al. studied chemically (bisphenol A-BPA and genistein-GEN) induced models of autism and revealed that BPA increased hippocampal and hypothalamic miR-153 and hippocampal miR-9 levels and decreased hypothalamic miR-181a levels, while GEN decreased hippocampal miR-7a and miR-153 expression. Hippocampal miR-9 and miR-153 expression and hypothalamic miR-181a and miR-153 expression were significantly correlated with the observed social deficits [56]. Furthermore, the deletion of miR-301a led to the emergence of anxiety, cognitive, and social deficits, which are considered autism-like behaviors in mice, while the exogenous administration of a miR-301a inhibitor reduced maternal immune activation (MIA)-induced autism-like behaviors probably through the upregulation of *SOCS3*, which inhibits the release of proinflammatory cytokines [59]. These findings provide preliminary yet important indications that miR-301a could be a promising therapeutic target in the clinical management of autistic phenotypes. Likewise, in another immune-mediated autism model, maternal immune activation transiently altered the expression of small RNAs, such as miRNAs, tRNA halves, and tRNA-derived small fragments (tRFs) at the maternal-fetal interface [63].

Human studies have confirmed that several miRNAs are differentially expressed in the blood, saliva, or faeces of individuals with autism and intellectual disability [25,29,30,32,40,41,44,45,47], and these expression patterns are associated with age and symptom severity. Moreover, structural variations of miRNAs and miRNA-target regions have been identified in patients with ADHD [35,53] and autism and could potentially disrupt the complementary base pairing in the miRNA-mRNA complex, given that they are seated in the mature region of miRNAs or in conserved 3’ UTR miRNA target sites [53].

According to relevant findings, the most upregulated miRNAs in autistic individuals were hsa-miR-302b-3p, hsa-miR-302a-3p, hsa-miR-302d-3p, hsa-miR-181a-5p, hsa-miR-155-5p, hsa-miR-3135b, hsa-miR-657, hsa-miR-2110, hsa-mir-4700, hsa-miR-191-5p, hsa-miR-139-5p, hsa-miR-432-5p, hsa-miR-665, hsa-miR-4705, hsa-miR-620, hsa-miR-1277-5p, hsa-miR-151a-3p, hsa-miR-125a-5p, hsa-miR-28-3p, hsa-miRNA 197-5p, hsa-miRNA-664a-3p, hsa-miR-424-5p, hsa-miR-7-5p and hsa-miR-2467-5p [25,29,30,32,40,41,45,52]. Likewise, the most down-regulated miRNAs in autistic individuals were hsa-miR-4433b-3p, hsa-miR-221-3p, hsa-miR-23a-5p, hsa-miR-4755-3p, hsa-mir-937, hsa-mir-3197, hsa-mir-103a-1, hsa-miR-28-3p, hsa-miR-148a-5p, hsa-miR-151a-3p, hsa-miR-125b-2-3p, and hsa-miR-7706, hsa-miR-500a-5p, hsa-miR-23a-3p, hsa-miR-32-5p, hsa-miR-140-3p, hsa-miR-628-5p and hsa-miR-484 [25,30,32,34,40,41,45]. It should be noted that hsa-miR-151a-3p was found to be either upregulated [29,52] or downregulated [30] in autism, and this discrepancy could be attributed to the samples’ age and phenotypic differences. In line with that, it was observed that the miR-302 family (hsa-miR-302a-5p, hsa-miR-302c-3p, hsa-miR-302a-3p, hsa-miR-302d-3p, hsa-miR-302b-3p, hsa-miR-302c-5p and hsa-miR-302b-5p) and hsa-miR-135b-5p were expressed at significantly higher levels in individuals with more severe autism symptoms, while hsa-miR-532-5p, hsa-miR-15b-3p, hsa-miR-30c-5p and hsa-miR-484 levels were lower in severe ASD cases [34,40,41]. Moreover, miRNAs expression was associated with the type of autistic symptomatology and the presence of co-morbidity [30]. In addition, miRNAs levels significantly correlated with age, mostly in ASD children and to a lesser extent in typically developing children [40].

The dysregulation of several miRNAs in neurodevelopmental disorders suggested that these molecules could be used as diagnostic and prognostic biomarkers. For example, Salloum-Asfar et al. [40] (2025) identified a set of 4 miRNAs (hsa-miR-4433b-5p, hsa-miR-15a-5p, hsa-miR-335-5p and hsa-miR-1180-3p) which exhibited high diagnostic sensitivity (83%) and specificity (97%) for the detection of autism, while Hicks et al. (2023) [30] showed that a set of hsa-miR-151-a-3p, hsa-miR-148a-5p, and hsa-miR-125b-2-3p could accurately identify ASD children. According to a more recent study, the hsa-miR-151a-3p can accurately predict neurodevelopmental delay in at-risk infants [29], while a combination of two miRNAs (hsa-miR-500a-5p and hsa-miR-197-5p) manifested high diagnostic power for autism spectrum disorder [32]. In a similar way, hsa-miR-7i-3p and hsa-miR-409-5p predicted the emergence of autism or intellectual disability in individuals with tuberous sclerosis [44], while another set of 5 miRNAs (hsa-miR-7-5p, hsa-miR-23a-3p, hsa-miR-32-5p, hsa-miR-140-3p, hsa-miR-3529-3p) exhibited high sensitivity and specificity to discriminate between children with any developmental delay and children with typical development [45]. Finally, the combination of hsa-miR-484 and its competing lncRNA *H19* showed high diagnostic value for ASD [34].

### 3.4. Small Nucleolar RNAs

SnoRNAs are a class of mid-size, between 60 and 300 nucleotides in length, conserved housekeeping noncoding RNAs that contribute to ribosome biogenesis [41,79]. They interact with their respective core proteins and enzymes and form snoRNA ribonucleoprotein (snoRNP) complexes, which catalyse specific modifications on target ribosomal RNA and small nuclear RNAs [80,81]. In addition, research has revealed that snoRNAs may also interact with transfer RNAs (tRNAs) and mRNAs [82,83]. SnoRNAs safeguard the structural and functional integrity of ribosomes, thus guaranteeing the accuracy and efficiency of protein synthesis and, in this respect, play a crucial part in cellular processes [82,83]. Recent research has focused on their contribution to pathogenetic mechanisms underlying cancer, cardiovascular diseases, and neurodegenerative diseases; in addition, there is sparse but emerging data linking snoRNAs to impaired neurodevelopment [41,84].

The current review included a small series of human studies that have assessed the role of snoRNAs in autism and Prader-Willi syndrome [24,28,41,51]. Salloum-Asfar et al. (2021) [41] detected 9 snoRNAs that were upregulated and 4 snoRNAs that were downregulated in the blood of ASD children compared to healthy controls. In addition, the expression level of certain snoRNAs was associated with the severity of autistic symptoms, with snoRA69 (also known as *U69*) being the most upregulated snoRNA and snoRD42A (*U42*) being the most downregulated snoRNA in children with severe autism. In another investigation, a molecular signature for the accurate diagnosis of autism was determined, which encompassed 20 circulating ncRNAs, including a snoRNA, the *U105B* [24]. Research has also shown that another snoRNA, the snoRD116, is critically implicated in the pathogenesis of Prader-Willi, which is a genetic syndrome with autism-like symptomatology [28,51]. This molecule belongs to the class of box C/D snoRNAs, which guide the 2’-O-methylation of precursor ribosomal and small nuclear RNAs within the nucleolus by base-pairing [28].

### 3.5. Circular RNAs

CircRNAs represent another class of long, single-stranded non-coding RNAs that are highly expressed in brain tissues. They are generated through the covalent joining of back-spliced exons; more specifically, the 5’-donor site of an intron is paired with the 3’-acceptor site of the targeted intron, forming a lariat as a by-product from the splicing event. CircRNAs’ circular structure confers great stability, and they are differentially expressed depending on neuronal activity and developmental stage [62,85]. CircRNAs interact with RNA-binding proteins, seclude miRNAs, and modulate gene expression through transcriptional control, and they play an important part in synaptic activity [58]. In addition, apart from their regulatory functions, certain circRNAs may encode peptides, thus having multifaceted contributions to intracellular processes [74]. Laboratory animal models have shown that circRNAs are probably involved in autism and fetal alcohol spectrum disorder [58,62]. *CircCaca1a*, *circCacna1c*, *circHivep2*, *circCdh9*, *circCdc14b*, *circTrpc6*, *circCep112*, *circWdr49*, and *circNcoa2* were found to be downregulated, while *circZcchc11*, *circRmst*, and *circMyrip* were found to be upregulated in the hippocampus of a mouse model of autism. In addition, it was revealed that *circCdh9* was also significantly downregulated in the cerebellum and upregulated in the prefrontal cortex and the amygdala of autism animal models, while both *CDH9* and *TRPC6* genes have been implicated in autism pathogenesis in humans [58]. Similarly, an earlier animal study reported that prenatal alcohol exposure altered circRNA expression in the fetal brain in a sex-dependent manner and extinguished the majority of the sex-specific changes in circRNA expression that physiologically occur in the typically developing brain. Among these differentially expressed circRNAs were *CircPtchd2* and *CircSatb2*, which were particularly increased in male fetal brains after alcohol exposure, and the latter was found to be strongly associated with inhibitory and excitatory neuronal gene expression [62].

### 3.6. Miscellaneous Classes of ncRNAs

Other classes of ncRNAs, including piRNAs, snRNAs, rRNAs, tiRNAs, and tRFs, have also been implicated in gene regulation at a transcriptional or translational level and seem to contribute to cell growth, differentiation, and function [12]. PiRNAs are a group of small, 24–31 nucleotides long, non-coding RNAs which were originally detected in reproductive cells; however, there is accumulating data that are expressed in several organs, including the brain [85]. They are typically generated within the nucleus as single-stranded 5′ monophosphorylated piRNA precursors and then are exported to the mitochondria, where they enter primary and ping-pong piRNA pathways [85]. Although their function has not yet been fully elucidated, they seem to directly interact with PIWI proteins to form piRNA-induced silencing complexes (piRISC), which regulate lncRNAs and mRNAs in their 3′ untranslated region [41,86]. Two human studies reported that several piRNAs are either down- or upregulated in the feces and plasma of ASD children [25,41]. The top three downregulated in ASD were hsa-piR-6691, hsa-piR-6693, and hsa-piR-29205, while the most upregulated were hsa-piR-28269, hsa-piR-32987, and hsa-piR-28059 in autistic children compared to their neurotypical siblings [25], while in another study, the most upregulated piRNAs were piR-hsa-1282 and piR-hsa-12790, and the most significantly downregulated were piR-hsa-32159 and piR-hsa-32167 [41]. Differential piRNAs expression levels were also detected between severe and mild ASD cases, with piR-hsa-22380, piR-hsa-28131, piR-hsa-27134, and piR-hsa-27138 being the most upregulated piRNAs and piR-hsa-27623 and piR-hsa-32175 being the most downregulated piRNAs in children with severe autism [41].

SnRNAs are small RNA molecules with a length of 100-300 nucleotides. They interact with proteins to form spliceosomes that play a major part in splicing RNA by removing introns from pre-mRNAs and transforming them into mature mRNAs. Spliceosomes constitute crucial regulators of gene expression, thus contributing to normal development and homeostasis, and the dysregulation of snRNAs may trigger pathogenetic intracellular processes [85]. In a large cohort of over 8800 individuals with genetically undiagnosed neurodevelopmental disorders, it was revealed that specific variants in a region of the *RNU4-2* gene, which encodes the U4 snRNA component of the major spliceosome, were associated with the emergence of severe neurodevelopmental deficits. These variants were exclusively observed in the maternal allele and could explain the presence of neurodevelopmental disorder in 0.4% of the cohort. In that same study, variants in other snRNAs genes were not significantly associated with neurodevelopmental phenotypes [23]. In addition, a group of snRNAs, namely *RNU1-16P*, *RNU6-258P*, *RNU6-485P*, *RNU6-549P* and *RNVU1-15*, were included in a blood-derived 20-ncRNA diagnostic signature which exhibited very satisfactory diagnostic accuracy for autism [24].

RRNAs are the most abundant RNA molecules in eukaryotic cells, and their function is to bind to proteins in order to form the small and large subunits of the ribosomes and recognize motifs in tRNAs and mRNAs. Six types of human rRNAs have been described: four (5S, 5.8S, 18S, and 28S) encoded by the nuclear genome and two (12S and 16S) encoded by the mitochondrial genome. Novel RNA sequencing technology has revealed that rRNAs are frequently cleaved into small non-coding fragments named short non-coding RNAs (srRNAs) whose role in cellular pathology is currently beginning to unravel [87]. A recent in vitro human study of Fetal Alcohol Spectrum Disorder showed that alcohol exposure downregulated 18S rRNA expression and decreased the levels of 18S rRNA-derived small fragments (srRNAs) in fetal neurons, while the exogenous administration of srRNAs reversed alcohol effects by increasing the expression of 18S rRNA and enhancing cell survival. Moreover, alcohol-induced reduction in srRNAs levels was strongly correlated with fetal eye size, which is a hallmark of Fetal Alcohol Spectrum Disorder, further emphasizing the potential contribution of srRNAs in its pathogenesis [26]. Finally, a non-coding rRNA, the *RNA5SP160* transcript, was included in the 20-ncRNA diagnostic signature for autism [41].

Another class of small ncRNAs is tRNA-derived RNA fragments (tRFs) and tRNA halves (tiRNAs), which are derived from tRNA or pre-tRNA and are involved in the regulation of protein synthesis [15]. According to a study of an immune-mediated autism model, maternal immune activation transiently altered the expression of tRNA halves and tRFs at the maternal-fetal interface, and more specifically, a set of 5ʹ tRNA halves were down-regulated, while a set of 18-nucleotide *tRF-3a* were up-regulated [63].

## 4. Discussion

The present review integrated a significant amount of the latest research findings, which emphasize the contribution of non-coding RNAs in the pathophysiology of various neurodevelopmental disorders. Existing reviews on this topic either report findings dating back to the period prior to 2020 or have a rather narrow focus as far as the type of neurodevelopmental disorder or the class of the studied ncRNAs is concerned [18,19,20,21,22]. In contrast, our review includes studies focusing on a wide range of neurodevelopmental conditions, including autism, ADHD, intellectual disability, Prader-Willi syndrome, fetal alcohol spectrum disorders, and specific learning disorders. Moreover, we have reviewed studies that assessed the role of multiple categories of ncRNAs, including housekeeping and regulatory RNAs, and have revealed that these RNA molecules are involved in numerous gene pathways, which underline immune-related and neuronal pathophysiological processes. According to our findings, most research has focused on the role of lncRNAs and miRNAS in neurodevelopment and attempted to delineate the underlying molecular events; however, there were also novel data regarding the effects of less-studied classes of ncRNAS, such as circRNAs, snoRNAs, piRNAs, tiRNAs, tRFs, etc. Among the neurophysiological pathways which are considered crucial for normal neurodevelopment, ncRNAs were found to regulate neuronal proliferation, apoptosis, migration and differentiation, synaptic plasticity, axonal growth and guidance, dendritic growth and density, and neurotransmitter signaling.

Animal studies and to a lesser extent human studies have consistently shown that lncRNAS, miRNAs and circRNAS are differentially expressed in the hippocampus, hypothalamus, cerebellum, striatum, prefrontal cortex, amygdala, the placenta, fetal brain, neural progenitor cells and pluripotent cells of animal models and human subjects with neurodevelopmental disorders, mostly autism, ADHD, learning disability and fetal alcohol spectrum disorders [26,33,35,46,47,54,55,56,57,58,59,60,61,62,63,64]. Anatomical alterations in these brain areas, including atypical patterns of cortical thickness, surface area, and gyrification, have been found in autistic and other neurodiverse populations [88,89] and have been linked to the clinical severity of the observed deficits in social interaction, communication, and sensory processing. Moreover, there are emerging findings linking these atypies in brain morphology to specific cellular processes. For example, increased cortical thickness in autistic children has been attributed to deviant migration and pruning resulting in excess neurons, increased neuronal size, greater dendritic spine density, and increased number of synapses. In a similar way, impairment in white matter development and synaptic pruning deficits may also be involved in the atypical expansion of the cortical surface observed in autism [90].

In addition, research has shed light on the molecular mechanisms that underlie the aberrant neuronal architecture and the disturbed signaling that lies at the core of neurodevelopmental disorders. Autism-related brain morphology and its corresponding pathophysiological processes are regulated through genetic and epigenetic influences [91]. As far as genetic factors are concerned, atypical brain development has been associated with variations in genes encoding chromatin modifiers and transcriptional regulators (e.g., *MECP2*, *CHD8*, *MEF2C*) and synaptic genes encoding neurexins (e.g., *NRXN2* and *NRXN3*), neuroligins (e.g., *NLGN3*, *NLGN4*), synaptic scaffolding proteins (e.g., *SHANK2* and *SHANK3*), synaptic receptors and neurotransmitter-related proteins (e.g., *GABRB3*, *GluR6*), and ion channels (e.g., *SCN2A*, *CACNA1D*, *CACNA2D3*) [89,92].

In this context, the current review highlights findings suggesting that several of the aforementioned susceptibility genes are associated with specific ncRNAs. Zarantonello et al. (2021) reported that the *CHD8* gene regulates the expression of the SINE-UP ncRNA *RAB11B-AS1*, which is indirectly involved in membrane and vesicle trafficking and apical protein recycling in the brain via the enhancement of translation of its sense counterpart, the protein-coding gene, *RAB11B* [54]. In addition, the exogenous administration of synthetic SINE-UP *CHD8* lncRNA molecules was shown to promote the production of the Chd8 protein, thus reverting the neurodevelopmental deficits caused by *CHD8* knock-down [65]. Similarly, the lncRNA *LINC00461* and its mouse homologue *NDIME*, which were found to be associated with autism and ADHD [35,55], regulate the expression of the *MEF2C* gene, which encodes a transcription factor, the myocyte-specific enhancer factor 2C [35,93]. *MEF2C* is highly expressed in the brain and plays a key role in neurogenesis [94] by inhibiting neuronal cell apoptosis [95], enhancing neuronal synapse formation [96,97], and affecting neural progenitor cell differentiation and maturation [98]. Another study, which provided evidence regarding the association between ncRNAs and autism risk genes, revealed that the administration of exosomal lncRNAs, including the lncRNA *IFNG-AS1*, in a mouse model of autism restored normal levels of *SHANK2*, *SHANK3*, and *MECP2* mRNAs, thus ameliorating neuroinflammation and promoting neurogenesis [57]. SHANKs are major multidomain scaffold proteins found at postsynaptic densities (PSDs) of excitatory synapses, which are linked to cell adhesion molecules, are involved in actin remodeling, and overall play a crucial role in synaptic formation and plasticity [99,100,101]. Mutations of the *SHANK* family genes (*SHANK2* and *SHANK3*) have been associated with autism pathophysiology [100]. Another gene, which is considered the primary cause of Rett syndrome, a rare yet debilitating neurodevelopmental disorder, and is also associated with increased autism risk, is the *MeCP2* gene, which encodes the methyl-CpG-binding protein 2. Mutations in the *MeCP2* gene contribute to aberrant dendritic morphology, which compromises synaptic maturation and function [102]. According to our review, a recent human study reported that a set of miRNAs, namely hsa-miR-106a-5p, hsa-miR-181a-5p, hsa-miR-195-5p, and hsa-miR- 328-3p, which were significantly upregulated in autistic individuals, targeted the *MECP2* gene [52].

Neurexins and neuroligins constitute two families of plasma membrane proteins that have been increasingly implicated in autism pathophysiology. More specifically, neurexins are presynaptic plasma membrane proteins that act as cell adhesion molecules and modulate synaptic functions. They form signaling complexes with neuroligins, which are postsynaptic plasma membrane proteins, and these complexes modulate the function of glutamergic and GABAergic synapses, thus maintaining the balance between excitatory and inhibitory transmission [103,104]. According to a recent investigation, the deletion of sno-LncRNAs and SPA-lncRNAs in human induced pluripotent cells led to the downregulation of *NRXN1* and *NRLG1* genes, which are involved in synaptogenesis and synaptic plasticity, and this regulatory effect could explain the role of those lnRNAs in neurodevelopment [46].

Another major finding of the current review was that several clusters of ncRNAs, mainly lncRNAs, miRNAs, piRNAS, and snoRNAs, might serve as reliable biomarkers for the diagnosis of autism and possibly the classification of its severity level and the presence of co-morbid conditions [24,25,29,30,32,34,40,41,42,45,48,49,50]. Current clinical practice in the field of neurodevelopmental disorders is based on caregiver- and teacher-reported information, clinical observation, and the completion of subjective scales and questionnaires, and lacks objective biomarkers for early and accurate diagnosis [30,105]. Although we strongly believe that clinical observation and multi-informant reports should remain in the core of the diagnostic algorithm, the use of highly sensitive and specific molecular indices might contribute to more cost-effective screening, earlier diagnosis, more accurate prognosis, and even tailor-made and more effective management.

Finally, a small number of recent animal and in vitro studies have reported promising results regarding the therapeutic potential of certain ncRNAS such as the lncRNAs *NDIME* and *IFNG-AS1* [55,57], the synthetic antisense lncRNA SINEUP-CHD8 [65] and a miR-301a inhibitor [59] for autism and a group of short rRNAs for fetal alcohol spectrum disorders [26]. Although this evidence originates from preclinical research and there is still a huge distance to cross in order to reach the point of testing RNA-based therapeutic regimens in clinical trials, a novel field in the medical management of neurodevelopmental disorders is currently emerging that, in the future, might complement existing psychosocial and educational interventions, thus optimizing treatment outcomes.

## 5. Conclusions

In conclusion, recent research has corroborated and expanded current knowledge regarding the contribution of non-coding RNAs to neurodevelopmental outcomes both in animal models and humans. Most relevant studies have focused on autism and on the categories of long ncRNAs and miRNAs; however, there is emerging evidence on the role of novel and not fully investigated classes of ncRNAs, such as circRNAs, piRNAs, and snoRNAs. A large number of ncRNAs are differentially expressed in neurodevelopmental conditions, and expression levels seem to be associated with the severity of the observed deficits. In line with that, certain RNA molecules such as the *LINC00461*, the *HOTAIR*, the hsa-miR-151a-3p, and the *SNORD116* have been consistently shown to be associated with specific neurodevelopmental conditions, including autism, ADHD, and Prader-Willi syndrome. In addition, several classes of ncRNAs, mostly lncRNAs, miRNAs, piRNAs, and snoRNAs, have appeared as promising diagnostic markers for autism with high sensitivity and specificity. These ncRNAs are abundant in human tissues, exhibit tissue-specific expression, are stable and easily detectable in various extracellular biomaterials including serum, plasma, saliva, cerebrospinal fluid, urine, and faeces, and due to these characteristics, they constitute ideal candidates for the accurate and non-invasive detection of neurodevelopmental conditions [14,38,39]. Moreover, there is limited yet extremely interesting data regarding the therapeutic potential of ncRNAs for neurodevelopmental deficits. RNA-based therapeutic approaches, including miRNA mimics or inhibitors (antisense oligonucleotides), are currently being developed in various clinical fields such as cancer, cardiovascular diseases, and infections [106]. According to the current review, three animals, one in vitro study, and one combined animal and in vitro study have shown that the administration of lncRNAs, srRNAs, a synthetic antisense lncRNA, and a miRNA inhibitor may promote neurogenesis, inhibit neuroinflammation, and ameliorate developmental deficits associated with autism and FASD [26,55,57,59,65]. However, these results are quite preliminary and need further corroboration. A limitation of the current systematic review is the absence of a formal risk of bias assessment, due to the heterogeneity of study designs (human, animal, and in vitro) and the descriptive nature of the included data, which precluded the use of standardized tools for bias assessment.

Figure 3 provides a comprehensive overview of the transcriptional origin, classification, and functional roles of coding and non-coding RNAs, highlighting their regulatory impact and potential involvement in neurodevelopmental disorders.

As this field advances, ncRNAs hold promise not only as molecular markers but also as potential mediators of neuroplasticity and therapeutic modulation. Their involvement in neurogenesis, synaptic integrity, and gene network regulation positions them as central players in the evolving landscape of neuropsychiatric research. The integration of transcriptomic data into clinical frameworks could redefine diagnostic criteria and treatment strategies, moving beyond symptom-targeted approaches to mechanism-based interventions. Future studies should focus on longitudinal profiling of ncRNA expression, functional validation in disease models, and the refinement of delivery systems for RNA-based therapeutics. Moreover, a unified methodological framework for risk of bias assessment in mixed-design studies would strengthen the evidence base. By addressing these gaps, future research can build on the foundation laid by this review, contributing to the realization of precision medicine in the management of neurodevelopmental disorders.

In total, the current review summarizes valuable scientific knowledge in a rapidly evolving area, which has the potential to revolutionize the understanding of neurodevelopmental processes at a molecular level, thus opening new paths in the prevention and management of neurodevelopmental aberrations and deficits. In the era of precision medicine, transcriptome analysis and network analysis studies might provide useful data for the discovery of novel drugs that target specific cellular and molecular mechanisms, thus obtaining more precise and clinically efficacious therapeutic benefits compared to traditional existing pharmacological agents, which only target symptoms [107]. According to this review, important original data have emerged; however, further research is needed to clarify the role of ncRNAs in neurodevelopment, shed more light on the underlying molecular and physiological mechanisms, assess their prognostic and diagnostic value, and design and implement potential RNA-mediated therapeutic regimens targeting core neurodevelopmental deficits.

## Figures and Tables

**Figure 1 biomedicines-13-01808-f001:**
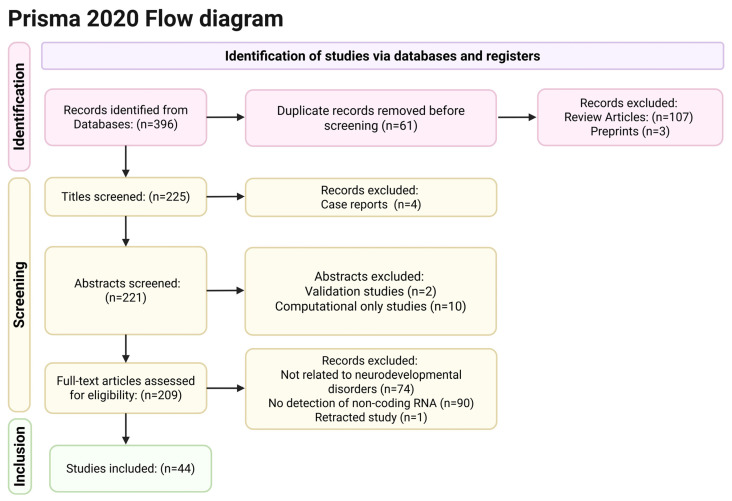
Flow Diagram for Article Selection Process. Created according to PRISMA guidelines using BioRender.com (accessed on 18 June 2025).

**Figure 2 biomedicines-13-01808-f002:**
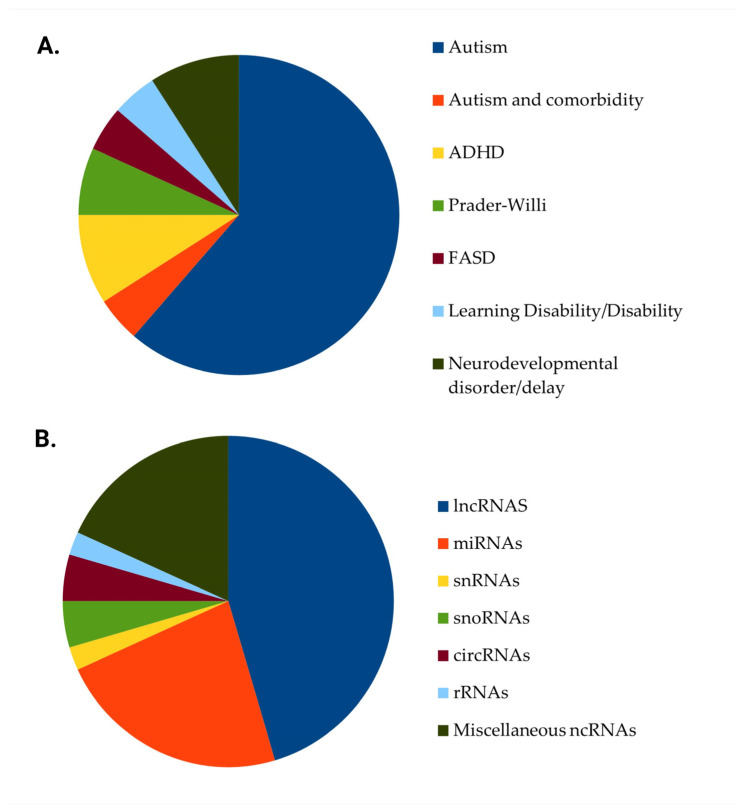
Distribution of studies based on neurodevelopmental conditions and ncRNA class. (**A**) Bar chart summarizing the distribution of studies by neurodevelopmental disorders. (**B**) Bar chart summarizing the distribution of studies by class of non-coding RNAs (ncRNAs). Abbreviations: ADHD, Attention-Deficit/Hyperactivity Disorder; FASD; Fetal Alcohol Spectrum Disorders; lncRNA, Long non-coding RNA; miRNA, MicroRNA; snRNA, Small nuclear RNA; snoRNA, Small nucleolar RNA; circRNA, Circular RNA; rRNA, Ribosomal RNA; ASD, Autism Spectrum Disorder.

**Figure 3 biomedicines-13-01808-f003:**
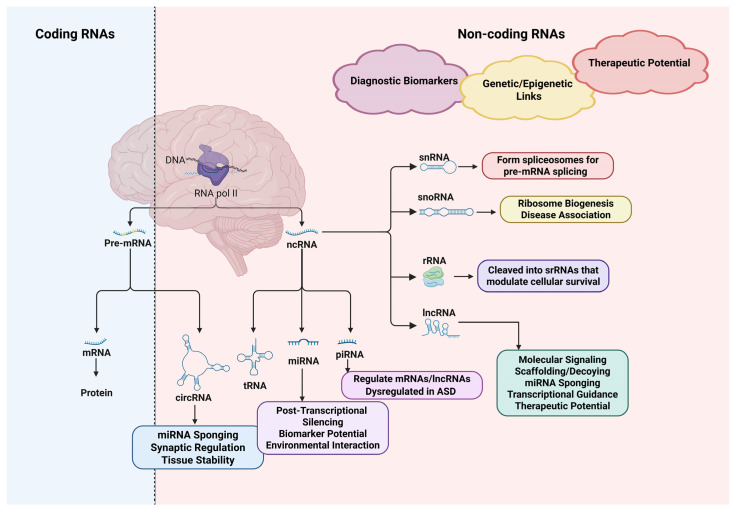
Illustration of the transcription and functional divergence of coding and non-coding RNAs, with a focus on their regulatory roles in neurodevelopmental disorders. RNA polymerase II transcribes DNA into precursor mRNAs (pre-mRNAs) and non-coding RNAs (ncRNAs). Pre-mRNAs are spliced into messenger RNAs (mRNAs), which are translated into proteins. In contrast, ncRNAs do not code for proteins but exert diverse regulatory functions. These include microRNAs (miRNAs), circular RNAs (circRNAs), PIWI-interacting RNAs (piRNAs), long non-coding RNAs (lncRNAs), transfer RNAs (tRNAs), small nuclear RNAs (snRNAs), small nucleolar RNAs (snoRNAs), and ribosomal RNAs (rRNAs). Non-coding RNAs influence gene expression through mechanisms such as splicing (snRNAs), ribosome biogenesis (snoRNAs), RNA stability and synaptic regulation (circRNAs), and post-transcriptional silencing (miRNAs and piRNAs). They also act as scaffolds, decoys, and transcriptional guides (lncRNAs), or are processed into small regulatory RNAs (srRNAs) that impact cell survival (rRNAs). These molecules are implicated in the pathophysiology of neurodevelopmental disorders nd exhibit potential as diagnostic biomarkers and therapeutic targets due to their genetic and epigenetic regulatory capacity. Created with BioRender.com (accessed on 10 June 2025). Abbreviations: RNA pol II, RNA polymerase II; Pre-mRNA, Precursor messenger RNA; mRNA, Messenger RNA; ncRNA, Non-coding RNA; miRNA, MicroRNA; piRNA, PIWI-interacting RNA; lncRNA, Long non-coding RNA; circRNA, Circular RNA; tRNA, Transfer RNA; snRNA, Small nuclear RNA; snoRNA, Small nucleolar RNA; rRNA, Ribosomal RNA; srRNA, Small regulatory RNA; ASD, Autism Spectrum Disorder.

**Table 1 biomedicines-13-01808-t001:** Characteristics of studies reporting on the role of non-coding RNAs in neurodevelopmental disorders.

Author	Ref. No	Date	Country	Type of Study	Disorder	Type of ncRNAs	Dysregulated ncRNAs	Biological Function	Biomaterial
Bai M et al.	[55]	2020	China	Animal	Autism	lncRNAs	*NDIME*	Neural differentiation	Hippocampus
Butler MC et al.	[56]	2020	USA	Animal	Socio-communicative deficits	miRNAs	miR-153, miR-181a, miR-9, miR-7a	Memory, neurons´ proliferation and differentiation	Hippocampus, Hypothalamus
Chen Y et al.	[23]	2024	UK, USA, Australia	Human	Neurodevelopmental disorder	snRNAs	*U4 snRNA*	-	Blood
Cheng W et al.	[24]	2020	China	Human	Autism	lncRNAs, pseudogenes, miscRNAs, rRNAs, snoRNAs, snRNAs	*AQP4-AS1*, *FTH1P2*, *FTH1P3*, *GMCL2*, *HMGN2P11*, *IGHV3-47*, *MUC20-OT1*, *MYCLP1*, *POLR2KP2*, *RN7SL132P*, *RNA5SP160*, *RNU105B*, *RNU1–16P*, *RNU6-258P*, *RNU6-485P*, *RNU6-549P*, *RNVU1-15*	-	Blood
Chiappori F et al.	[25]	2022	Italy	Human	Autism	miRNAS and piRNAs	Hsa-miR-657, hsa-miR-2110, hsa-piR-28059	Cell-cell junction, metabolite signalling, bacterial invasion, inflammation	Faeces
Darbinian N et al.	[26]	2023	USA	Human	Fetal Alcohol Spectrum Disorders	small ribosomal RNAs	srRNA 10, srRNA 11, srRNA 14, srRNA 47	Neural cell proliferation, migration and apoptosis	Fetal brain tissue
Duan K et al.	[27]	2023	USA, Netherlands	Human	ADHD	lncRNAs	*RP11_6N13.1*	-	Blood
Fu Y et al.	[57]	2024	China	Animal	Autism	lncRNAs	*IFNG-AS1*	Neurogenesis, inhibition of apoptosis	Prefrontal cortex
Gasparini S et al.	[58]	2020	Italy	Animal	Autism	circRNAs	*CircCdh9*, *circCacna1a*, *circCacna1c*, *circHivep2*, *circCdc14b*, *circTrpc6*, *circCep112*, *circWdr49*, *circNcoa2*, *circZcchc11*, *circRmst*, *circMyrip*	Neuron development, glutamate receptor signaling, synaptogenesis, synaptic plasticity	Hippocampus
Heimdorfer D et al.	[28]	2024	Austria, Germany	Human	Prader-Willi, Schaaf-Yang	snoRNAs	*SNORD116*		Cell lines
Hicks SD et al.	[29]	2023	USA	Human	Neurodevelopmental delay	miRNAs	miR-125a-5p, miR-148a-5p, miR-151a-3p, miR-28-3p	Inflammatory pathways, neuron apoptosis	Saliva
Hicks SD et al.	[30]	2020	USA	Human	Autism	miRNAs	miR-28-3p, miR-148a-5p, miR-151a-3p, miR-125b-2-3p, miR-7706, miR-665, miR-4705, miR-620, miR-1277-5p	Axonal guidance, neurotrophic signaling, GABAergic synapse, addiction pathways	Saliva
Isik CM et al.	[31]	2025	Turkey	Human	Specific learning disorder	lncRNA	*SYNGAP1-AS1*	Synaptic plasticity, learning, long-term potentiation	Blood
Kichukova T et al.	[32]	2021	Bulgaria	Human	Autism	miRNAs	miR-500a-5p, miR-197-5p, miR-424-5p, and miR-664a-3p	Synaptic pathways, axon guidance, neuroactive ligand-receptor interaction	Serum
di Leva F et al.	[65]	2025	Italy, France, USA	Ιn vitro human and animal cells, in vivo animal model	Autism	antisense long ncRNA	SINEUP-*CHD8*	Regulation of cell proliferation and migration	Cell lines
Li D et al.	[33]	2021	USA	Human	Autism	lncRNAs	*DNM3OS*, *IGFBP7-AS1* and *LINC01139*	Axonal sprouting	Neural progenitor cells, neurons
Li X et al.	[59]	2024	China	Animal	Autism	miRNA-301a	miR-301a	Cytokine pathways Synaptogenesis signaling pathway	Brain tissue
Li Y et al.	[34]	2024	China	Human	Autism	lncRNAs, miRNAs	*H19*, miR-484	Postsynaptic density, presynaptic active zone, synaptic membrane, nucleobase-containing compound kinase activity, and regulation of RNA stability	Blood
Liu Y et al.	[35]	2020	USA	Human	ADHD	miRNAs, lncRNAs	miR137 Long ncRNA *LINC00461*	Neuroactive ligand-receptor interaction pathway	Blood
Liu S et al.	[35]	2020	China	Animal	Autism and ADHD	lncRNAs	*LINC00461*	Neural migration, neural differentiation	Hippocampus
Liu H et al.	[60]	2024		Animal	Learning disability	lncRNAs	gi|672032535, uc.447, gi|672048769, NON-RATT015986, NONRATT027328, NONRATT022126, gi|672015918, gi|672027113gi|672071577, NON-RATT016007, NONRATT028784, uc.447, NONRATT022126, gi|672015918, NONRATT016007	Pancreatic secretion pathway	Hippocampus
Mendes M et al.	[36]	2025	Canada	Human	Autism	lncRNA	*DDX53/PTCHD1-AS*	**-**	
Mizuno S et al.	[61]	2020	Japan	Animal	Autism	ncRNAs	LncRNAs *Gm26917* and *Gm37194*	Synaptic transmission, neuron signalling, immune signalling	Cerebral cortex, striatum
Paudel P et al.	[62]	2020	USA	Animal	Fetal Alcohol Spectrum Disorder	circRNAs	*CircSatb2*, *circPtchd2*	GABA receptor signaling, semaphoring signaling, dopamine signaling, neuron development, neuritogenesis, axonal guidance	Brain
Piras IS et al.	[37]	2021	USA, Italy	Human	Autism	lncRNAs	*MALAT*-1	Synaptogenesis, dendritic density	Blood
Rahmani Z et al.	[38]	2024	Iran	Human	Autism	lncRNAs	*DISC2*, *Linc00945*, *Foxg1-as1*, *Csnk1a1p*, *Evf2*	Neuron differentiation, migration	Blood
Safari M et al.	[39]	2020	Iran	Human	Autism	lncRNA	*HOTAIR*	Immune-mediated pathways	Blood
Salloum-Asfar S et al.	[40]	2025	Qatar	Human	Autism	miRNAs	miR-4433b-5p, miR-151a-5p, miR-335-5p, and miR-1180-3p	Neural differentiation and apoptosis, synaptogenesis, immune-related pathways	Plasma
Salloum-Asfar S et al.	[41]	2021	Qatar	Human	Autism	miRNAs, piRNAs, snoRNAs,	hsa-miR-302b-3p, hsa-miR-302a-3p, hsa-miR-302d-3p, piR-hsa-1282, piR-hsa-12790, *SNORD3C*, *SNORD69*, *SNORD51*	Neural development Synaptic plasticity Gut homeostasis	Plasma
Sane S et al.	[42]	2024	Iran	Human	Autism	lncRNAs	LincRNA-ROR, LINC-PINT, LincRNA-p21, *PCAT-29*, *PCAT-1 l*		Blood
Sayad A et al.	[43]	2020	Iran	Human	ADHD	lncRNA	*HOTAIR*	Neuron apoptosis	Blood
Scheper M et al.	[44]	2022	Netherlands, Italy, Germany, Austria, Belgium, Australia, France, UK, USA, Poland, Czech Republic	Human	Autism, Intellectual Disability	miRNAs and isomiRNAs	hsa-miR-409-5p, hsa-miR-1301-3p, hsa-miR-145-5p, hsa-miR-412-5p, hsa-miR-423-3p, hsa-miR-154-5p, hsa-miR-214-5p, hsa-miR-376b-3p, hsa-miR-379-3p, hsa-miR-494-3p, hsa-miR-103a-3p, hsa-miR-410-3p_miRNA, hsa-miR-221-3p_trim3	Neurotransmitter signaling, neuroplasticity	Serum
Sehovic E et al.	[45]	2020	Bosnia Herzegovina	Human	Neurodevelopmental delay/disorder	miRNAs	miR-7-5p, miR-23a-3p, miR-27a-3p, miR-32-5p, miR-140-3p, miR-628-5p, miR-2467-5p	Regulation of cell proliferation and differentiation	Saliva
Sledziowska M et al.	[46]	2023	UK	Human	Prader Willi	spa-lncRNAs, sno-lncRNAs	*SNHG14*, *RPL4*, *RPS17*, *SNURF-SNRPN*, *PWAR6*, *IPW*, *SPA1*, *SPA2*, sno lncRNA1, sno lncRNA2, sno lncRNA3, sno lncRNA4, sno lncRNA5	Neuron polarity, synaptogenesis, synaptic transmission, disturbed proliferation, increased apoptosis, cell-cell signalling	Human induced pluripotent cells
Su Z et al.	[63]	2020	USA	Animal	Autism	tRFs, miRNAs	5ʹ halves from tRNAAsp, tRNAGly, tRNAGlu, tRNAVal, tRNAiMet, tRNALeu tRNASeC and tRNACys, 5ʹ halves from tRNAGly and tRNAGlu, tRF-3Tyr, tRF-3Gln, tRF-3Thr, tRF-3Leu, tRF-3Ser, tRF-3Trp and tRF-3Ala, miR-291a/b-5p, Mmu-miR-146b-5p, Mmu-miR-215-5p,	Cytokine pathways, embryo development	Mouse placenta/decidua
Sun JJ et al.	[47]	2022	China	Human	Autism	lncRNAs, miRNAs	49 lncRNAs, 30 miRNAs	Mitochondrial-energy metabolism, cellular communication, transcriptional regulation, and glial-cell fate,	Brain tissue, blood
Taheri M et al.	[48]	2021	Iran	Human	Autism	lncRNAs	*CCAT1* and *CCAT2*	Immune-related pathways, T cell function	Blood
Taheri M et al.	[49]	2021	Iran	Human	Autism	lncRNAS	*MEG3*	Neuronal synaptic plasticity, neuronal apoptosis	Blood
Tamizkar KH et al.	[50]	2021	Iran	Human	Autism	lncRNAs	*DISC2*, *PRKAR2A-AS1*, *LRRC2-AS1*, *LOC101928237*	Dopaminergic activity, cerebral hypoperfusion	Blood
Tan Q et al.	[51]	2020	Canada	Human	Prader-Willi	snoRNAs	*SNORD116* cluster	-	Blood
Wang Z et al.	[52]	2022	China	Human	Autism	miRNAs	hsa-miR-191-5p, hsa-miR-151a-3p, hsa-miR139-5p, hsa-miR-181a-5p, hsa-miR-432-5p	Dendritic spine formation, axon growth, neural differentiation, synaptogenesis	Blood
Wong A et al.	[53]	2022	USA	Human	Autism	miRNAs	Hsa-miR-6780a-3p, hsa-miR-1225-5p, hsa-miR-2277-3p, hsa-miR-548j-5p, hsa-miR-100-5p	Regulation of neuronal death, dendritic spine formation, early forebrain dorsal-ventral pattern formation	Blood
Zarantonello G et al.	[54]	2021	Italy, USA	In vitro, human cells	Autism	antisense lncRNAs	*RAB11B-AS1*	Vesicular trafficking, synaptic activity	Human induced neural progenitor cells
Zhang S et al.	[64]	2020	China	Animal	ADHD	lncRNAs	NONRATT027852.2, NONRATT005132.2, NONRATT004982.2 NONRATT027914.2, NON-RATT006598.2, NONRATT002035.2, NON-RATT016515.2	Synaptic plasticity regulation	Hippocampus

ADHD: Attention Deficit Hyperactivity Disorder; ncRNAs: non-coding RNAs, lncRNAs: long non-coding RNAs, miRNAs: microRNAs, snoRNAs: small nucleolar RNAs, snRNAs: small nuclear RNAs, circRNAs: circular RNAs, piRNAs: Piwi-interacting RNAs, rRNAs: ribosomal RNAs, tRFs: tRNA-derived small fragments.

## Data Availability

The data that support the findings of this study are available from the first author upon reasonable request.

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
