# Peer review of "Non-Coding RNAs in Neurodevelopmental Disorders—From Diagnostic Biomarkers to Therapeutic Targets: A Systematic Review"

_biomedicines, 2025, doi:10.3390/biomedicines13081808_

Round 1

Reviewer 1 Report

Comments and Suggestions for Authors

A brief summary

Interest in non-coding nucleic acids has been growing lately. This is related to the fact that, in addition to regulating transcription, translation, and other processes of genetic information implementation, non-coding nucleic acids represent a specific regulatory mechanism based on interference. It is noteworthy that the review discusses lncRNAs, miRNAs, snRNAs, snoRNAs, circRNAs, and rRNAs, which is very valuable, as considering the involvement of such acids in the regulation of the development of nervous system pathologies provides an opportunity to look at another level of regulation and formulate new ideas regarding the development of new drugs in the future. Out of 76 literature sources, 64 sources were published after 2020. No self-citation excessive number were found.

General concept comments

  1. Why is the SFARI database specifically mentioned in line 36? Is it a specialized database containing specific data about autism? Or was it chosen randomly? It would be beneficial to add a statement justifying the mention of this database.
  2. The article would carry much more weight with the inclusion of information about the partners with which the mentioned non-coding nucleic acids from Table 1 interact. For example, for miRNAs, one can find information in databases about which genes they regulate. If there are many genes (potentially more than 1000) regulated by a single miRNA, a cluster analysis for GO:BP (biological processes) could be performed. If, for example, it is lncRNA, additional information about which miRNAs it can bind to, acting as a sponge, would be helpful. This review might interest not only scientists but also physicians, for whom the analysis of miRNAs seems complex; if genes are listed in the table, understanding will be much more effective and useful. This suggestion is left to the authors' discretion, as it is not straightforward and could be considered as recommendations for future work.
  3. In the section dedicated to miRNAs (line 228), it is necessary to add information at the beginning stating that most information about miRNAs is only known for those with numbers up to 1000. Therefore, the function and role of the miRNAs collected from publications do not always mean that the role of these miRNAs in the development of nervous system disorders is known.
  4. In Figure 3, therapeutic potential is mentioned. Before the 'Conclusions' section, it would be useful to include information about existing or developing drugs targeting these non-coding nucleic acids. For example, there is a common technology of antisense nucleotides being developed to bind with ncRNAs. This remark, like the second one, can be perceived by the authors as a recommendation for continuing the work.

Specific comments

  1. In the paragraph starting on line 259, miRNAs are listed, but not all have prefixes indicating the species, while the remaining part has the prefix hsa-. The same applies to line 406.

Author Response

Reviewer 1

Comment #1: “Why is the SFARI database specifically mentioned in line 36? Is it a specialized database containing specific data about autism? Or was it chosen randomly? It would be beneficial to add a statement justifying the mention of this database.”

Response: Thank you for your comment. A brief statement regarding the SFARI database has been added to the text (lines 45-46).

Comment #2: “The article would carry much more weight with the inclusion of information about the partners with which the mentioned non-coding nucleic acids from Table 1 interact. For example, for miRNAs, one can find information in databases about which genes they regulate. If there are many genes (potentially more than 1000) regulated by a single miRNA, a cluster analysis for GO:BP (biological processes) could be performed. If, for example, it is lncRNA, additional information about which miRNAs it can bind to, acting as a sponge, would be helpful. This review might interest not only scientists but also physicians, for whom the analysis of miRNAs seems complex; if genes are listed in the table, understanding will be much more effective and useful. This suggestion is left to the authors' discretion, as it is not straightforward and could be considered as recommendations for future work.”

Response: Indeed, this is a very insightful comment. Most manuscripts report a plethora of genes regulated by the studied ncRNAs. We have included in our manuscript some examples of genes that are regulated by certain ncRNAs (the lncRNA NDIME regulates the transcription of the MEF2C gene) and of miRNAs that are inhibited by specific long ncRNAS ( LINC00461 acts as sponge to miRNA-411-5p), however, this is a large amount of information that could not be easily integrated in the current review. We thank the reviewer for his/her suggestion, and we will take it into serious consideration for future work.

Comment #3: “In the section dedicated to miRNAs (line 228), it is necessary to add information at the beginning stating that most information about miRNAs is only known for those with numbers up to 1000. Therefore, the function and role of the miRNAs collected from publications do not always mean that the role of these miRNAs in the development of nervous system disorders is known.”

Response: Thank you for your useful suggestion. The information has been added (lines 292-296).

Comment #4: “In Figure 3, therapeutic potential is mentioned. Before the 'Conclusions' section, it would be useful to include information about existing or developing drugs targeting these non-coding nucleic acids. For example, there is a common technology of antisense nucleotides being developed to bind with ncRNAs. This remark, like the second one, can be perceived by the authors as a recommendation for continuing the work”

Response: Thank you for your suggestion. We have included the required information in the manuscript (lines 481-490) and we also take the reviewer´s recommendation into consideration for future work.

Specific comment 1: In the paragraph starting on line 259, miRNAs are listed, but not all have prefixes indicating the species, while the remaining part has the prefix hsa-. The same applies to line 406.

Response: Thank you for your comment. It has been corrected throughout the manuscript

Reviewer 2 Report

Comments and Suggestions for Authors

The manuscript entitled "Neurodevelopmental disorders and non-coding RNAs: insights from recent advances" is a review of papers from 2020 to 2025. The authors described their selected logics and tabulated the discovery of these papers. It provides quick information for the experts in this field.

I have two suggestions to improve Table 1: (1) Authors might include the first name and add "et al," such as Bai, M. et al. (2) Type of study might mention the specific experiments, such as transcriptome analysis or others. I do not understand what 'experimental' and 'human' mean.

Author Response

Reviewer 2

Comment #1: “Authors might include the first name and add "et al," such as Bai, M. et al. ”

Response: Thank you for your comment.  It has been done.

Comment #2: “Type of study might mention the specific experiments, such as transcriptome analysis or others. I do not understand what 'experimental' and 'human' mean.”

Response: The reviewer is right. We have replaced the word experimental with the word animal which refers to laboratory animal studies, while humans refer to studies with human participants.

Reviewer 3 Report

Comments and Suggestions for Authors

Karaivazoglou et al. aims to provide insights from recent advances in neurodevelopmental diseases and non-coding RNAs. To this end, the authors search Pubmed and ScienceDirect on March 23, 2025 using the phrases “non-coding RNAs and neurodevelopmental disorders” and “non-coding RNAs and autism”. Such a search yielded 396 titles, which was filtered down to 44 research articles. Then the authors summarized the findings from these articles, particularly focusing on long non-coding RNAs (lncRNAs), microRNAs (miRNAs), circular RNAs (circRNAs), small nucleolar RNAs (snoRNAs) and other miscallenous RNAs. Presentation of key data in a table makes it easy for readers to follow the text. Finally, the authors complete the manuscript with a succinct conlusion.

Although many genetic defects have been associated with neurodevelopmental disorders that affect communication, social interaction, sensory processing and behaviors of individuals, the contribution of non-coding RNAs is partially understood. The authors provide an up-to-date account of current research articles on neurodevelopmental diseases and non-coding RNAs. I believe that this review would benefit the researches working in the field. I recommend the following points for consideration to improve the manuscript.

  1. Line 15-16: “translational level” should be “post-transcriptional level”
  2. Line 44: It would be nice to divide non-coding RNAs into two categories first, namely housekeeping ncRNAs and regulatory ncRNAs (e.g. PMID: 35657049). I suggest focusing merely on regulatory RNAs, eliminating the parts on structural or housekeeping RNAs as I will point out in the following items.
  3. Figure 2: Please label each pie chart “A” and “B” and provide a legend.
  4. Lines 146-147: The authors nicely describe the molecular functions of lncRNAs. It would be nice to state that a small fraction of lncRNAs are translated into micropeptides. I wonder whether there are any lncRNA-translated micropeptides with functions in neurodevelopmental diseases.
  5. Lines 284-319: I recommend removing “small nucleolar RNAs” part as they are housekeeping RNAs.
  6. Line 327: Some circular RNAs are translated. Please insert that statement. I wonder if there are any circRNA-translated proteins with functions in neurodevelopmental diseases.
  7. Line 344: Under the subtitle “Miscellaneous classes of ncRNAs”, I would recommend deleting the parts related to snRNAs and rRNAs; and instead, include studies on tRFs or tsRNAs.

Minor points:

  1. Line 32: “is” should be “are”

Author Response

 Reviewer 3

Comment #1: “Line 15-16: “translational level” should be “post-transcriptional level”

Response: Thank you for your comment.  It has been done.

Comment #2: “Line 44: It would be nice to divide non-coding RNAs into two categories first, namely housekeeping ncRNAs and regulatory ncRNAs (e.g. PMID: 35657049). I suggest focusing merely on regulatory RNAs, eliminating the parts on structural or housekeeping RNAs as I will point out in the following items.”

Response: Thank you for your constructive comment. We added the required information regarding the two categories of ncRNAs in the introduction section (lines 71-80). However, we respectfully opt to retain coverage of certain housekeeping RNAs (e.g., snRNAs, snoRNAs, rRNAs), as there is increasing evidence that these classes, beyond their canonical roles, exert regulatory effects relevant to neurodevelopmental disorders. Thus, despite their traditional classification, these RNAs have regulatory and pathophysiological significance in neurodevelopmental contexts, justifying their inclusion in a comprehensive review on the topic.

Comment #3: “Figure 2: Please label each pie chart “A” and “B” and provide a legend.”

Response: Thank you for your comment. It has been done.

Comment #4: “Lines 146-147: The authors nicely describe the molecular functions of lncRNAs. It would be nice to state that a small fraction of lncRNAs are translated into micropeptides. I wonder whether there are any lncRNA-translated micropeptides with functions in neurodevelopmental diseases”

Response: Thank you for your suggestion. We have added that statement (lines 197-200). The reviewer poses a very interesting question that could be addressed in future work.

Comment #5: “Lines 284-319: I recommend removing “small nucleolar RNAs” part as they are housekeeping RNAs.”

Response: Thank you for your comment. We understand the reviewer’s rationale, given the classical categorization of snoRNAs as housekeeping RNAs involved in rRNA modification. However, we respectfully believe that removing this section would omit key insights. Recent research demonstrates that certain snoRNAs are differentially expressed in neurodevelopmental conditions such as autism and Prader-Willi syndrome. These findings support the emerging notion that snoRNAs may act beyond ribosome biogenesis, possibly affecting neural differentiation and brain development. 

Comment #6: “Line 327: Some circular RNAs are translated. Please insert that statement. I wonder if there are any circRNA-translated proteins with functions in neurodevelopmental diseases.”

Response: Thank you for your constructive remark. We have added that statement (lines 387-389). Your question regarding the potential role of circRNA-translated proteins in neurodevelopment raises a highly interesting issue that could addressed in future work.

Comment #7: “Line 344: Under the subtitle “Miscellaneous classes of ncRNAs”, I would recommend deleting the parts related to snRNAs and rRNAs; and instead, include studies on tRFs or tsRNAs.”

Response: Thank you for your comment. We have included a study on tRFs and tRNA halves in that section (lines 457-463). However, we respectfully suggest not omitting the snRNA and rRNA sections, as both are supported by recent high-impact studies demonstrating their relevance. Thus, we respectfully argue that these ncRNA classes deserve representation in a review aiming to reflect the broad and evolving scope of RNA-mediated regulation in neurodevelopmental disorders.

Minor points

Line 32: “is” should be “are”

Response: The reviewer is right. It has been corrected.